# Surgical Interventions in Cases of Esophageal Hiatal Hernias among Older Japanese Adults: A Systematic Review

**DOI:** 10.3390/medicina58020279

**Published:** 2022-02-13

**Authors:** Yuta Horinishi, Kai Shimizu, Chiaki Sano, Ryuichi Ohta

**Affiliations:** 1Matsue Seikyo General Hospital, 8-8-8 Nishituda, Matsue 690-8522, Shimane, Japan; yuuta881219@yahoo.co.jp; 2Department of Community Medicine Management, Faculty of Medicine, Shimane University, Izumo 693-8501, Shimane, Japan; sanochi@med.shimane-u.ac.jp; 3Huchu Hospital, 1-10-17, Huchu-Town, Izumi 594-0076, Osaka, Japan; lkaaishimi7738@gmail.com; 4Community Care, Unnan City Hospital, 96-1 Iida, Daito-cho, Unnan 699-1221, Shimane, Japan

**Keywords:** esophageal hiatal hernia, surgery, Japan, older adults, ageism

## Abstract

*Background and Objectives*: Given Japan’s superaging population, an increasing number of older adults in the country need surgical treatment for esophageal hiatal hernias. Accordingly, this systematic review examines surgical interventions for symptomatic esophageal hiatal hernias in older Japanese patients and explores treatment outcomes. *Materials and Methods*: Articles on single operations for hiatal hernias published after 1991 were found on Google Scholar and Ichushi using specific keywords. Subsequently, articles fulfilling the predetermined inclusion criteria were considered in the study. *Results*: The mean patient age was 81.4 years, and the male-to-female ratio was 1:11.5. The main reasons for surgery were vomiting, dyspnea, and chest tightness. In terms of hernia classification, type IV was the most common (48%). Surgical modalities were laparoscopy in 15 cases, and laparotomy in 10 cases. Mean postoperative course was 26.47 days until hospital discharge, and there were no cases of perioperative death. *Conclusions*: Findings showed that multiple factors were involved in older adults’ prognoses, and age was not the only biological factor. Therefore, aggressive surgical intervention should be considered for symptomatic older patients, even in the absence of surgery indicators.

## 1. Introduction

A hiatal hernia is the most common type of diaphragmatic hernia and is a relatively common disease, especially among older adults. This condition refers to the intra-abdominal organs overhanging into the thoracic cavity. According to a national survey conducted by the Society for Gastroesophageal Reflux Disease in Japan, hiatal hernias occur in 49.3% of individuals who undergo upper endoscopies, and are more common in men and older adults. With the increase in global population aging, the prevalence of this disease is also increasing [1] for reasons such as weakening of the diaphragmatic esophageal ligament, dilatation of the esophageal hiatus due to vertebral body fracture, gibbus deformity, obesity, and an increase in abdominal pressure [2]. There are four types of hiatal hernias: sliding (type I), pure paraesophageal (type II), mixed (type III), and complex (type IV) [3]. Type I is the most common (approximately 95% of the cases), while the prevalence of each of the other types is under 1% [4]. Hiatal hernias can cause several problems. The most common symptom of type I hernia is gastroesophageal reflux disease. Types III and IV, which are most common in older women, can cause dysphagia, passage disorder, anemia, strangulation, and aspiration pneumonia. Among the cases of esophageal hiatal hernia, relatively few cause respiratory and circulatory disorders [5]. Nevertheless, cases requiring emergency surgery include those where patients exhibit an incarcerated hernia, a prolapse, a gastric axis twist, a respiratory disorder due to lung compression [6], and arrhythmia due to cardiac compression [7]. Therefore, careful consideration is required when deciding the course of treatment for esophageal hiatal hernias in older adults.

Japan is one of the most rapidly aging countries in the world; accordingly, the number of surgical procedures performed in older adults is also increasing [8]. However, surgical treatments for older patients remain controversial. In general, surgical outcomes are expressed as a balance between the beneficial effects on the patient and the risk of death or sequelae. Therefore, surgical interventions are limited to cases with lower chances of mortality and fewer possible complications. Nevertheless, surgical treatment decisions for older cancer patients, especially those with comorbidities, appear to be primarily based on the surgeon’s subjective judgment [9]. Additionally, as a result of discrimination against older adults (ageism), these patients are sometimes not given the option of surgical procedures that are considered to be the standard for younger patients [8,10]. Previous studies showed strong correlation between ageism, and physical and psychological risks in older adults [11,12,13]. Prognosis for older adults is the result of a combination of multiple factors; thus, prognosis may be improved by determining these factors and intervening as appropriate. According to the Guidelines for the Management of Hiatal Hernia [14], surgical interventions are strongly recommended for patients with type II–IV hernias, acute obstruction, and an axial twist. Clinically, cases of obstruction and axial torsion are often handled by surgical interventions because of their urgency. However, hernia types II–IV in older patients may not be operated on, even though it is strongly recommended. Ageism may play a role in this decision.

Furthermore, there is limited research on surgery for older patients with hiatal hernias. Therefore, this study focused on Japan’s superaging society to examine surgical interventions for symptomatic esophageal hiatal hernias in older patients and explore treatment outcomes. Our study results may help physicians and surgeons in deciding whether standard surgical interventions should be used for older patients who had been conservatively treated solely because of their age.

## 2. Materials and Methods

### 2.1. Research Methods

This systematic review is prepared according to the preferred reporting items for systematic reviews and meta-analyses (PRISMA) guidelines. The study flow diagram is reported in Figure 1. The study was registered on the international prospective register of systematic reviews (PROSPERO)platform. Registration number: 283059.

### 2.2. Search Strategy

We searched two online databases, Google Scholar and Ichushi, using the following keywords: “shokudourekkouherunia” (esophageal hiatal hernia) and “koureisya” (older adult). Japanese keywords were used, as when relevant keywords in English were used to search Google Scholar and PubMed, no results matching our inclusion criteria were found.

### 2.3. Inclusion and Exclusion Criteria

In our study, older adults were defined as individuals aged 65 or older. Articles published after 1991 on surgeries to treat esophageal hiatal hernias in older patients were included in this study. Conversely, articles published before 1991 and studies or cases involving emergency operations, gastric volvulus, malignancies, intestinal perforations, and shock vitals were excluded. These cases were excluded because malignancy is a prognostic factor in hiatal hernia surgery. Furthermore, cases in which the patients were in poor condition (i.e., shock vitality) were also excluded, as the surgical intervention was performed for life-saving purposes. Patients with gastric volvulus and gastrointestinal obstruction were excluded because surgery was recommended for these conditions on the basis of guidelines for esophageal hiatal hernias. Our inclusion and exclusion criteria are summarized in Table 1.

### 2.4. Data Extraction

Data extraction was performed by searching for keywords “shokudourekkouherunia” and “koureisya” on Google Scholar and Ichushi.

### 2.5. Analysis

Quantitative and qualitative data are presented as descriptive statistics. Collected data were then divided into topics. We used Microsoft Excel to analyze the gathered data.

## 3. Results

### 3.1. Search Results

No reviews or case summaries on esophageal hiatal hernia in older Japanese patients could be found. Therefore, we focused on and reviewed case reports. A total of 1933 studies that met the abovementioned inclusion criteria were obtained. Of these, 1736 were excluded, as their titles or abstracts were unrelated to the current study. Further, after reviewing the entire texts, 172 were excluded for the following reasons: 67 concerned gastric volvulus; 24, malignant tumors; 40, gastrointestinal bleeding, occlusion, perforation, black death, and entrapment; 11 Morgagni hernias; 7, congestive heart and liver failure; and 6, shock vitals. Lastly, 17 studies were deemed to be duplicates and excluded (Figure 1). In the end, a total of 22 references were identified, and 25 case reports were included in the final analysis (Table 2) [6,15,16,17,18,19,20,21,22,23,24,25,26,27,28,29,30,31,32,33,34,35].

### 3.2. Demographics

The mean age of the patients was 81.4 years with a variance of 67.36. The male-to-female ratio was 1:11.5. The most common reason for surgery was vomiting (48%), followed by dyspnea (28%) and chest pain (8%). Other symptoms were heartburn, fever, worsening heart failure, postprandial syncope, abnormal liver function, wheezing, weight loss, anorexia, abdominal pain, and constipation.

### 3.3. Medical History

Hypertension (36%) was the most common medical condition. Other conditions included asthma, dementia, heart failure, diabetes, stroke, reflux esophagitis, pulmonary embolism, aortic valve stenosis, hiatal hernia, lumbar compression fracture, rheumatoid arthritis, acute myocardial infarction, humeral fracture, cholelithiasis, and osteoarthritis.

### 3.4. Types of Esophageal Hiatal Hernias

Type IV hernias were the most common, accounting for 48% of the cases, followed by type III at 32%; type I, undescribed at 8%; and type II at 4%.

### 3.5. Surgical Form

There were 15 (60%) laparoscopic and 10 (40%) laparotomy cases. For fundoplication, 12 patients underwent Nissen repair, 6 Toupet repair, 2 Hill repair, and 6 only hernia repair.

### 3.6. Age-Adjusted Charlson Comorbidity Index

Past history was noted in 22 cases, and the age-adjusted Charlson Comorbidity Index (CCI) [36] could be calculated: 3 cases had CCI3, 7 cases had CCI4, 10 cases had CCI5, 1 case had CCI6, and 1 case had CCI7. The mean CCI was 4.54.

### 3.7. Outcome

For the 25 cases, mean postoperative course was 26.47 days (range of 7–108 days) until hospital discharge, and no cases of perioperative death were reported. Furthermore, no cases of postoperative recurrence were found. Postoperative complications were reported in 1 of 15 laparoscopic cases (6.7%) and 5 of 10 laparotomy cases (50%). The mean postoperative hospital discharge time was 11.9 days (range of 7–21) in 13 of 15 laparoscopic cases, while it was 73.8 days (range of 24–108 days) in 4 of 10 laparotomic cases. Regarding the year of publication, six cases of laparotomy were published between 1991 and 2000, one between 2001 and 2010, and three between 2011 and 2021. For laparoscopic cases, 3 were reported between 2001 and 2010, and 12 from 2011 to 2021; however, no cases were reported from 1991 to 2000. Almost no reports described postoperative quality of life, and as such, this aspect was not included in the data collection for this study.

### 3.8. Study Characteristics

This systematic review aimed to analyze surgical interventions for symptomatic esophageal hiatal hernias in older Japanese patients and to explore treatment outcomes. We identified 22 references, and 25 case reports were included in the final analysis.

## 4. Discussion

In this systematic review, we aimed to analyze surgical interventions for older Japanese patients with symptomatic esophageal hiatal hernia and to explore treatment outcomes. We identified 22 references, and 25 case reports were included in the final analysis.

Previous studies [37] reported 5.4–17% mortality rates for emergency surgery, and an average mortality rate of 1.38% (0–5.2%) for laparoscopic standby surgery. Although the mean age of patients in this study was particularly high at 81.4 years, the average time to postoperative discharge was 26.47 days, and no cases of perioperative death were reported. We found a standby surgery mortality rate of 0%, which is better than that reported in a previous study (1.38%) [37]. Postoperative prognosis was good for patients who had undergone surgery, even those who were at a more advanced age. Previous studies showed strong correlation between ageism, and physical and psychological risks in older adults [11,12,13,38]. This supports the possibility that multiple factors contribute to the prognosis of older adults, and that age is not the only relevant biological factor. The mean CCI of the 22 patients who could be considered for CCI in this study was 4.54; the relative risk of death after 10 years was 4.40 for CCI4, and 6.38 for CCI5 rather than CCI0. However, one drawback of CCI is that activities of daily living (ADL) are is not taken into account. ADL is a prognostic factor among the elderly, and an association between walking speed and survival was reported [39]. In this study, most of the case reports did not describe ADL, and the lack of information on ADL is a limitation of this study. It is crucial to determine the indication for surgery by considering ADL and the patient’s intention, even if the patient is at high risk because of predictors such as CCI. Consideration of prognostic factors and appropriate interventions may improve the prognosis of older patients with esophageal hiatal hernias. Clinicians also need to be aware of the unconscious bias of ageism.

In this study, 92% of the patients were women. Their reasons for undergoing surgery were dyspnea, vomiting, orbital pain, difficulty in food intake, and other complications such as repeated aspiration pneumonia and the exacerbation of reflux esophagitis. Types III and IV accounted for most of the hernia cases. Statistically, esophageal hiatal hernia is more common in men across all age groups [1]; however, in this study, most patients were women. The reason for this discrepancy may be that types III and IV esophageal hiatal hernias increase with age in women owing to postmenopausal osteoporosis-associated gibbus deformity and habitual or deformed posterior lumbar curvature [1].

Among the excluded records (*n* = 172), nine were emergency surgery cases, and six of these surgeries were on male patients. Emergency surgery cases are usually more common in men. Older men are less attentive to their health than their female counterparts are [40]; this finding may explain the high number of men who underwent emergency surgery after their symptoms had worsened. Furthermore, married men are more concerned about their health than those who are separated or widowed are. This may indicate the influence of marriage on men’s health. Men tend to be less cognizant of caring for people of the same gender, regardless of their relationship, and are more likely to care for people of the opposite gender (e.g., spouses or partners). This suggests that partners and spouses may influence the behavior of men who do not seek medical attention until their condition worsens. Connections with other people decrease with age [41], but men, in particular, are expected to continue their connections with individuals of the opposite gender.

Regarding surgical interventions, laparoscopic surgery was performed in 15 cases, while laparotomy was performed in 10. Postoperative complications were reported in 1 of 15 laparoscopic cases (6.7%) and 5 of 10 laparotomy cases (50%). The mean postoperative hospital discharge time was 11.9 days (range of 7–21 days) in 13 of 15 laparoscopic cases. In contrast, the mean postoperative hospital discharge time was 73.8 days (range of 24–108 days) in 4 of 10 laparotomic cases. Laparoscopic surgery was associated with shorter postoperative hospital stays than laparotomy surgery was. The Nissen and Toupet methods for open surgery were used in clinical practice; however, after the report of Dallemagne et al. [42] on laparoscopic surgery in 1991, this surgical method also became widespread in Japan. Important aspects of surgery include the return of prolapsed organs, closure of the esophageal hiatus, formation of the cardia to prevent regurgitation, and the fixation of twist prevention; however, numerous sutures are required, and the high recurrence rate was initially a problem [43]. Over the years, the feasibility and safety of laparoscopic surgery were reported [44,45]. There was no difference in long-term prognosis between laparoscopic surgery and open surgery [46,47]. Nonetheless, laparoscopic surgery appears to ensure quicker hospital discharge [48]. The study results also support this fact. Laparoscopic surgery is expected to be effective for older adults for several reasons, including minimal invasiveness, better cosmetic appearance for the patient, and the ability to secure the surgeon’s visual field. Regarding year of publication, six cases of laparotomy were published between 1991 and 2000, one between 2001 and 2010, and three between 2011 and 2021. Results also reflect an increase in the number of laparoscopic surgeries.

Despite the significance of the findings, this study had some limitations. First, only the Japanese literature was included in the study; thus, racial differences in the findings must be considered. Second, most of the inquiry targets were case reports; therefore, cases with surgical failures may not have been included.

## 5. Conclusions

Overall, results suggested that prognosis was good in the included cases, although the average age of the patients was 81.4 years. The surgical method appeared to positively influence patient outcomes. The average discharge time was 26.47 days, and no cases of perioperative death were found.

On the basis of the results, even if there is no obvious surgical indicator for esophageal hiatal hernia in older adults (e.g., axial torsion or acute obstructive symptoms), aggressive surgical interventions might still improve the quality of life of symptomatic patients. Therefore, clinicians need to be aware of their unconscious bias of ageism while treating older patients, and reach decisions related to surgical indications on the basis of the patient’s prognostic factors. Further collaboration between surgeons and general practitioners specializing in reaching such decisions helps in improving the quality of life of older adults in a rapidly aging society such as Japan. In conclusion, the best method for treating esophageal hiatal hernia in older adults should be determined with a consideration of multiple factors and not just age alone.

## Figures and Tables

**Figure 1 medicina-58-00279-f001:**
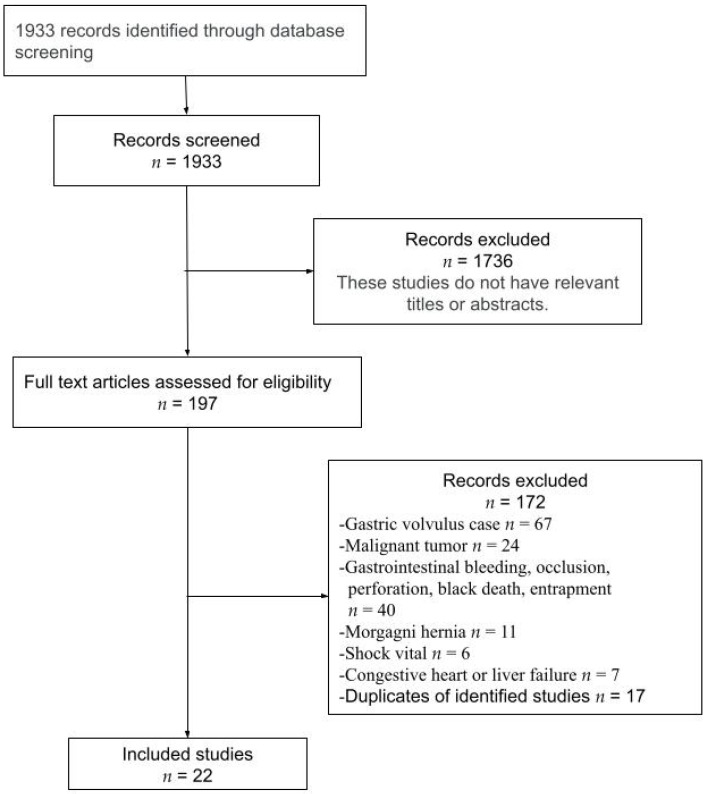
PRISMA flow diagram of the systematic searching process.

**Table 1 medicina-58-00279-t001:** Inclusion and exclusion criteria.

Inclusion Criteria	Exclusion Criteria
Articles published after 1991 on surgeries to treat esophageal hiatal hernias Patients aged 65 or older	Gastric volvulus Malignant tumors Gastrointestinal bleeding, occlusion, perforation, black death, and entrapment Morgagni hernias Congestive heart failure and liver failure Shock vitals Duplicates of identified studies

**Table 2 medicina-58-00279-t002:** Included articles.

Last Name of First Author (Year of Publication)	Patient’s Age	Patient’s Sex	Hernia Type	Reason for Surgery	Past History	Surgical Procedure Fundoplication	Postoperative Discharge	Mortality	Complications	Age-Adjusted Charlson ComorbidityIndex
Miyake et al. (1997) [6]	77	Female	I	Dyspnea	None	Laparotomy Nissen, Hill	Not stated	None	None	3
Yatabe et al. (2017) [15]	71	Female	IV	Dyspnea on exertion, heartburn	No remarkable findings	Laparoscopy Toupet	7 days	None	Ileus	3
Shikata et al. (2008) [16]	86	Female	IV	Palpitations, fever, cough	Chronic subdural hematoma, thigh bone fracture, dementia	Laparoscopy Nissen	10 days	None	None	5
Kawai et al. (2019) [17]	94	Female	III	Exacerbation of heart failure, dyspnea	Heart failure, pulmonary embolism, bronchial asthma	Laparoscopy Hernia repair	16 days	None	None	5
Saito et al. (2017) [18]	82	Female	IV	Repetitive postprandial syncope, cardiac compression due to hiatal hernia	Diabetes, anemia	Laparoscopy Hernia repair	Not stated	None	None	5
Yagi et al. (2018) [19]	81	Male	IV	Hepatic dysfunction	Hypertension, aortic stenosis	Laparoscopy Nissen	12 days	None	None	5
Okumura et al. (2015) [20]	71	Female	IV	Dyspnea, vomiting	Hypertension, uterine fibroids	Laparoscopy Nissen	16 days	None	None	4
Mimatsu et al. (2014) [21]	82	Female	II	Dysphagia	Hypertension, sinusoidal failure syndrome, cerebral infarction, dementia	Laparoscopy Nissen	Not stated	None	None	7
Masayuki et al. (2015) [22]	69	Male	III	Heartburn, dyspnea	Hypertension, kidney failure	Laparoscopy Toupet	11 days	None	None	5
Masayuki et al. (2015) [22]	81	Female	III	Vomiting	Asthma	Laparoscopy Toupet	21 days	None	None	4
Nonaka et al. (2013) [23]	86	Female	IV	Dyspnea, wheezing	Not stated	Laparotomy Nissen	About 90 days	None	Upper gastrointestinal obstruction	-
Ishino et al. (2012) [24]	79	Female	III	Weight loss, heartburn	Hypertension, hyperlipidemia	Laparoscopy Nissen	11 days	None	None	4
Okada et al. (2009) [25]	74	Female	III	Chest pain, vomiting	Asthma, GERD, hyperlipidemia, postoperative lung cancer, hypertension	Laparotomy Hernia repair	73 days	None	Pyothorax Pneumonia	4
Hirashita et al. (2008) [26]	93	Female	III	Vomiting, aspiration pneumonia	Hypertension	Laparoscopy Nissen	7 days	None	None	5
Hirashita et al. (2008) [26]	93	Female	III	Vomiting	None	Laparoscopy Nissen	7 days	None	None	4
Eda et al. (2000) [27]	79	Female	IV	Vomiting	Uterine myoma, diabetes mellitus, hypertension, chronic bronchitis	Laparotomy Hernia repair	24 days	None	Pneumonia	5
Matsui et al. (1992) [28]	87	Female	IV	Chest pain, multiple gastric ulcers	Lumbar spine compression fracture	Laparotomy Hill	Not stated	None	Pulmonary complications	4
Hirai et al. (2011) [29]	84	Female	IV	Aspiration pneumonia, vomiting, fever	Reflux esophagitis, chronic bronchiectasis	Laparoscopy Nissen	10 days	None	None	5
Uno et al. (2015) [30]	83	Female	IV	Vomiting	Hypertension, dementia	Laparoscopy Toupet	14 days	None	None	6
Kim et al. (2011) [31]	88	Female	III	Aspiration pneumonia, multiple gastric ulcers, vomiting, loss of appetite	Cerebral infarction, cataract	Laparotomy Hernia repair	Not stated	None	Not stated	5
Kajitani et al. (2016) [32]	67	Female	IV	Abdominal pain, constipation	Cesarean section, rheumatoid arthritis, acute myocardial infarction, humerus fracture	Laparotomy Toupet	108 days after reoperation	None	Septal vertical ulcer, subtransverse septal abscess, resurgery	3
Iyobe (1994) [33]	72	Female	Not stated	GERD, vomiting	Not stated	Laparotomy Nissen	Not stated	None	None	-
	76	Female	Not stated	GERD, vomiting	Not stated	Laparotomy Nissen	Not stated	None	None	-
Mori et al. (2011) [34]	100	Female	IV	Aspiration pneumonia, ARDS, vomiting blood	Gallstone, deformed spine, heart failure	Laparoscopy Toupet	13 days	None	None	5
Ishizakii et al. (1999) [35]	80	Female	I	Dyspnea on exertion	Asthma	Laparotomy Hernia repair	Not stated	None	None	4

GERD: gastroesophageal reflux disease; ARDS: acute respiratory distress syndrome.

## Data Availability

All relevant datasets in this study are presented in the manuscript.

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
