# Peer review of "Surgical Interventions in Cases of Esophageal Hiatal Hernias among Older Japanese Adults: A Systematic Review"

_medicina, 2022, doi:10.3390/medicina58020279_

Round 1

Reviewer 1 Report

I think that while supporting intervention on symptomatic patients, it may be a bit of a stretch to extend to all patients. One factor that might affect decision in frailty so you ight consider a comment on charlson score, age adjusted charlson score, modified frailty index or other measures of frailty.  You may have enough info to calculate these in your patients you describe and see how their outcomes compare to predicted. 

Author Response

We thank the reviewers for the valuable comments. Please find below our point-by-point responses to the comments.

â– Reply to reviewer 1's reports

Commented : I think that while supporting intervention on symptomatic patients, it may be a bit of a stretch to extend to all patients. One factor that might affect decision in frailty so you ight consider a comment on charlson score, age adjusted charlson score, modified frailty index or other measures of frailty.  You may have enough info to calculate these in your patients you describe and see how their outcomes compare to predicted.

Response:

Thank you for reviewing our paper. We agree with your feedback.

The mean age-adjusted Charlson Commodity Index (CCI) in our study was 4.54. One of the shortcomings of CCI is that it does not take ADL into account. ADL is known to be a prognostic factor when considering the elderly population. In this study, most of the case reports did not include ADL, and the lack of information on ADL is a limitation of this study. However, even if the patient is at high risk due to predictive factors such as CCI, it is extremely important to determine the indication for surgery by taking into account ADL and the patient's intentions. We have presently added these details in the revised version of our manuscript.

Reviewer 2 Report

“Surgical Interventions in Cases of Esophageal Hiatal Hernias Among Older Japanese Adults: A Systematic Review” by Yuta Horinishiet al describes the multiple factors involved in surgical interventions for symptomatic esophageal hiatal hernias in older Japanese patients. The data showed that aggressive surgical intervention should be considered for symptomatic older Japanese patients. The paper is well written in general. Moreover, there are several minor findings that should be pointed out.

Minor

  1. In the discussion line 176: the authors mention “92% of the patients were women”, the number of patients in this review was 24. Therefore, women patients are 20 patients and men patients are 4 patients? However, in lines 184-185, the authors described “Among the eligible cases, nine were emergency surgery cases, and six of these surgeries were on male patients”, please explain these issues.
  2. In the discussion lines 181-183: the authors mention “The reason for this discrepancy may be that types III and IV esophageal hiatal hernias increase with age in women owing to postmenopausal osteoporosis-associated gibbus deformity and habitual or deformed posterior lumbar curvature”, there are any research figured out the relationship between osteoporosis and esophageal hiatal hernias, aren’t there?

Author Response

We thank the reviewers for the valuable comments. Please find below our point-by-point responses to the comments.

â– Reply to reviewer 2's reports

  1. In the discussion line 176: the authors mention “92% of the patients were women”, the number of patients in this review was 24. Therefore, women patients are 20 patients and men patients are 4 patients? However, in lines 184-185, the authors described “Among the eligible cases, nine were emergency surgery cases, and six of these surgeries were on male patients”, please explain these issues.

Response: Thank you for bringing this to our attention. The number of patients included in this study is 25 instead of 24.There were two male patients and 23 female patients, leading to our statement that "92% of patients were female". Of the records excluded n=172, there were nine emergency surgeries. Of these, six were male. We have now clarified these details in our paper.

  1. In the discussion lines 181-183: the authors mention “The reason for this discrepancy may be that types III and IV esophageal hiatal hernias increase with age in women owing to postmenopausal osteoporosis-associated gibbus deformity and habitual or deformed posterior lumbar curvature”, there are any research figured out the relationship between osteoporosis and esophageal hiatal hernias, aren’t there?

Response: Thank you for your comment. We have added reference 1 to this study regarding the basis for the above statement.